

# Accuracy of numerical wave model results: Application to the Atlantic coasts of Europe

Matias Alday[1], Fabrice Ardhuin[1,2], Guillaume Dodet[1], and Mickael Accensi[1]

[1]Univ. Brest, CNRS, Ifremer, IRD, Laboratoire d'Océanographie Physique et Spatiale, Brest, France
[2]Scripps Institution of Oceanography, University of California San Diego, La Jolla, California

**Correspondence:** Matias Alday (malday@ifremer.fr)

**Abstract.** Numerical wave models are generally less accurate in the coastal ocean than offshore. It is generally suspected that a number of factors specific to coastal environments can be blamed for these larger model errors: complex shoreline and topography, relatively short fetches, combination of remote swells and local wind seas, less accurate wind fields, presence of strong currents, bottom friction, etc. These factors generally have strong local variations, making it all the more difficult to adapt

a particular model setup from one area to another. Here we investigate a wide range of modelling choices including forcing fields, spectral resolution and parameterizations of physical processes in a regional model that covers most of the Atlantic and North Sea coasts. We show that the accurate propagation of waves from offshore is probably the most important factor on exposed shorelines, while other specific effects can be important locally, including winds, currents and bottom friction.

## 1 Introduction

Numerical wave models have been used from the global ocean to the coast, for a wide range of applications, including the design and safe operation of sea-going structures such as ships, platforms and wind turbines. The progressive improvement of parameterizations in spectral wave models based on the wave action equation (WAE), like SWAN (Booij et al., 1999) or WAVEWATCH III®(The WAVEWATCH III® Development Group, 2019), has helped to continuously extend their use into coastal regions and areas with shallower water depths. With the introduction of currents, bottom friction related to different

sediment types and coastal reflection, errors in the main wave parameters have dropped to levels similar to open ocean simulations (Ardhuin et al., 2012; Roland and Ardhuin, 2014; Salmon et al., 2015). High resolution modeling has also become more efficient with the implementation of unstructured grids (mesh), providing flexible spatial resolution taking into account wave characteristics and bathymetry features (Benoit et al., 1996; Roland, 2008; Dietrich et al., 2011; Alves et al., 2013). In particular, previous works by Boudière et al. (2013) and Wu et al. (2020) present the implementation and validation of high

resolution hindcasts for wave resource assessments along French waters and the U.S West Coast respectively.

    In general, the accuracy of spectral models is a function of at least 3 main factors. First, the accuracy of forcing fields (e.g. Cavaleri and Bertotti, 1997), second, the realism of the parameterization of processes representing spectral wave evolution (e.g. Ardhuin et al., 2010) and third, the numerical choices made to integrate the WAE, namely discretization and numerical schemes (e.g. Tolman, 1995b; Roland and Ardhuin, 2014). For example, in the hindcast presented in Alday et al. (2021), more





accurate wave height distributions were obtained at global scale by adjusting parameterizations and discretizations. When it comes to nested models, the characteristics of the boundary conditions should also be taken into account.

In the present paper the analysis is extended to intermediate and shallow water depths. To this end, we present a high resolution wave hindcast for European Atlantic waters, using boundary conditions from Alday et al. (2021). Through out the study we attempt to determine which elements in the model setup have a significant effect on the characteristics of the simulated

sea states and hence the accuracy of the results. Given the wide range of bathymetry features, bottom sediment types, fetch and tidal amplitudes in coastal environments, we also verify when and where these choices introduce important changes.

Particular attention is paid to the effects of tidal currents, directional resolution and bottom friction over the simulated wave fields. Performance analysis of the results is conducted in terms of the significant wave heights, directional spreading, peak direction, and mean periods. Additionally, analyses on the energy distribution as a function of frequency were conducted to

further explore the changes introduced through modifications in the forcing, resolution or the boundary conditions.

Details on the model setup, source terms and numerical choices are presented in section 2. Wave measurements used for sensitivity analyses and validation in 3. The model performance analysis is described in section 4 followed by its validation and conclusions in sections 5 and 6.

## 2    Model setup and sensitivity tests

### 2.1    Mesh construction


The triangle grid used for the simulations was created using an interface developed at BGS IT&E. The main data sources employed for the mesh construction were coastline polygons from OpenStreetMap (last update of used data set: 2018-06-10 09:33), bathymetric information from EMODnet (2016 version) and HOMONIM digital terrain models (DTM). The DTM sources have gridded resolutions of ~210 and ~110 m respectively, with depths defined with respect to the mean sea level

(MSL). Although the coastline is generally located at high water levels with an exact definition that varies from country to country, we have chosen to impose a constant 2 m minimum depth value at the coastline to preserve the shoreline geometry and avoid unrealistic wave height gradients at the nearshore that could be triggered by the combination of large tidal sea level variations (wet and dry effect) with inadequate spatial resolution in very shallow areas close to the shore.

Previous to the triangulation, a nodes' homogenization of the coastlines was applied to ensure a minimum segment length

of 400 m in the polygons. An extra segment coarsening (up to 1200 m) and trimming was applied along the Norwegian fjords to reduce the final amount of nodes. This action facilitated obtaining a more relaxed CFL restriction, which implies a larger minimum time step for wave propagation, 13 s in this case, but it also implies that details of the Norwegian coastline are not as well resolved. In addition, nodes from an existing mesh (Boudière et al., 2013) with the exception of those placed less than 800 m from the coastlines, were included in the generation of the new mesh fixing their previous position. This was done to

facilitate the use of the new results by users of the previous hindcast.



Finally, the resolution was increased in 14 zones of interest for marine energy users, (Fig. 1a). The generated mesh has a total of 328030 nodes (Fig. 1b), with a resolution (triangle side) ranging from ∼200 m at the coast and refined zones to, approximately 15 km in deep offshore areas.

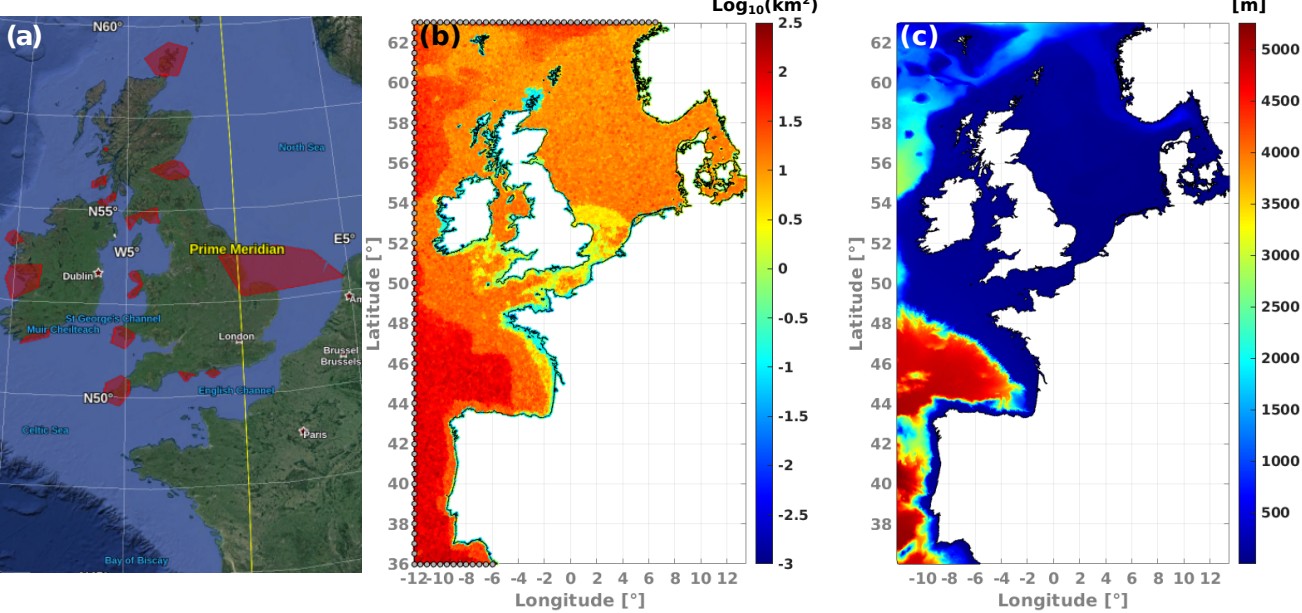

**Figure 1.** (a) Refinement polygons in red. (b) Final mesh elements size distribution, coastlines polygons in black, in grey mesh nodes where boundary conditions are prescribed from the global model. (c) Bathymetry reconstruction with mesh. Colorbar in (c) represents depths with respect to MSL in meters. Map data in (a) are from ©Google Landast / Copernicus.

## 2.2 Bottom sediment map

The construction of a sediment grain size map was included to properly represent wave energy dissipation due to bottom friction (see section 4.5 for results). In the model, the grain size is characterized by its median diameter $D_{50}$, defined at each node of the mesh. The $D_{50}$ values where estimated from the EMODnet harmonized seabed substrate charts. The minimum grain size was set to 0.02 mm, while zones characterized as boulders were represented with a $D_{50}$=150 mm. By default, the minimum grain size was applied to all regions where no substrate was specified. Since most areas with no bottom characterization are

in deep waters (e.g. > 400 m), this assumption does not have any relevant effect on the wave fields evolution. The bottom sediment diameter map is presented in Fig. 2.





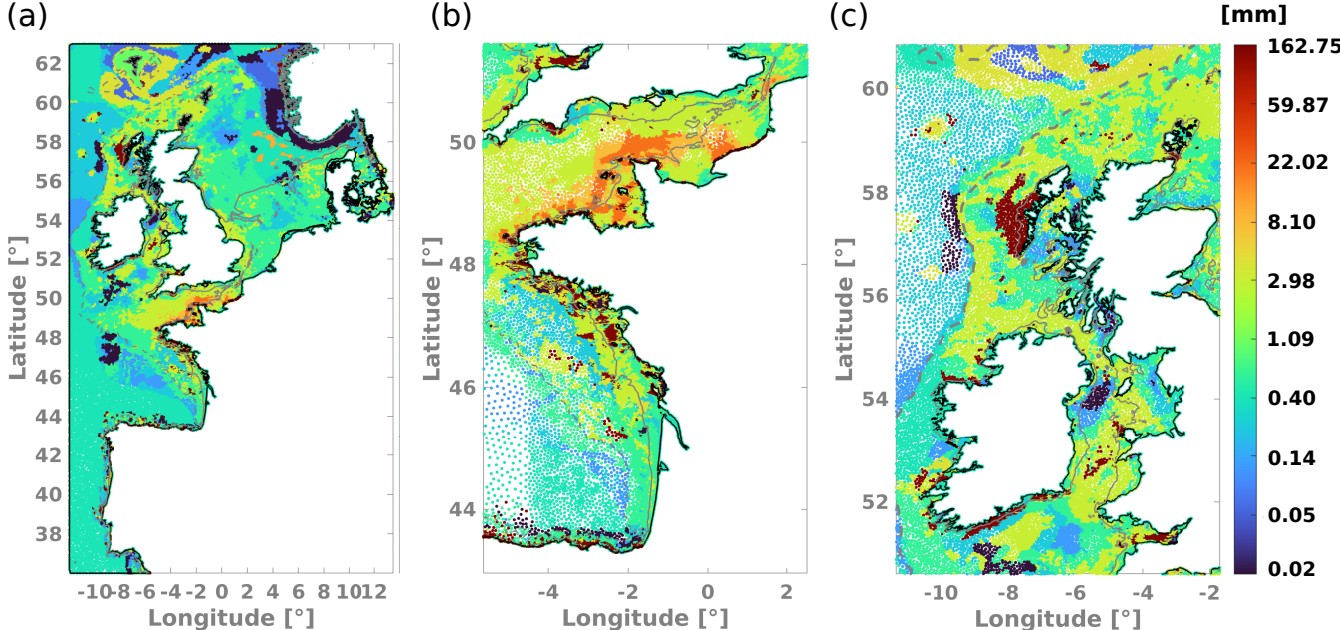

**Figure 2.** Bottom sediment size map. $D_{50}$ values assigned to each mesh node for: (a) Full domain, (b) Bay of Biscay and the English Channel and (c) UK. Colorbar represents $D_{50}$ in mm. Gray dashed lines represent 200 m depth contours, continuous gray lines represent 50 m depth contours.

## 2.3 Source terms and numerical choices

In WW3 the WAE is solved using a splitting method to treat in different steps temporal depth changes, spatial propagation, intra spectral propagation and source terms (Yanenko, 1971; Tolman and Booij, 1998; The WAVEWATCH III® Development

Group, 2019). Spectral propagation, which includes refraction, is computed with an explicit third order scheme that combines the QUICKEST scheme with the ULTIMATE total variance diminishing limiter (Leonard, 1991), while spatial advection is done with the explicit Narrow stencil scheme (N-scheme) (Csík et al., 2002; Roland and Ardhuin, 2014). Non linear evolution and wave to wave interactions are represented with the Discrete Interaction Approximation (DIA, Hasselmann et al., 1985). The utilized wind input and wave dissipation source terms are taken from the ST4 parameterizations described in Ardhuin et al.

(2010) with adjustments described in Alday et al. (2021) consistent with the global model used for our boundary conditions. A constant wave energy reflection of 5% is used at the coastlines, as parameterized by Ardhuin and Roland (2012).

In the present study we only analyze the effects of changes in the ST4 parameterizations. A detailed list of the parameters used for the model implementation is given in Appendix A.



### 2.4 Boundary conditions and forcing fields improvements

The accuracy of modelled wave data directly depends on the quality of the forcing fields and the provided boundary conditions (BC) for the case of nested models. This becomes particularly relevant in coastal areas, for accounting wave-current interactions in macro tidal areas, the assessment of energy resources, port design and operation conditions, or the study of extreme events.

Along with the high spatial resolution, an important aspect of the wave hindcast analyzed in this study, is the utilization of improved spectral BC from the wave data set described in Alday et al. (2021). This wave hincast was created using wind fields

from the fifth generation ECMWF atmospheric reanalyses of the global climate, ERA5 (Hersbach et al., 2020), and surface current fields taken from the CMEMS Global Ocean Multi Observation Products (MULTIOBS_GLO_PHY_REP_015_004).

The global grid from where boundary conditions are taken has a spatial resolution of 0.5°, while the wave spectrum is discretized in 24 directions (15° resolution) and 36 exponentially spaced frequencies from 0.034 to 0.95 Hz with a 1.1 increment factor from one frequency to the next. The proposed spectral discretization, wave growth and dissipation parameters, along

with the use of upgraded forcing fields, showed clear improvements of sea state parameters (at global scale) when compared to previous hindasts, like the widely used data set from Rascle and Ardhuin (2013).

The (directional) spectral BC taken from the global model are prescribed along the southern, western and northern open boundaries of the mesh (Fig. 1b). These are interpolated in space and time into each active node along the open boundaries of the nested model.

For the proposed regional model, three main forcing fields were included: wind, tidal levels and tidal currents. As for the global model, ERA5 surface winds were used for wave generation. Similar to what was done in Boudière et al. (2013), tidal levels and currents time series were reconstructed in WW3 with harmonics taken, in this case, from two different sources. The first one, is the output from Ifremer's tidal atlas (Pineau-Guillou, 2013) created with MARS 2D (Lazure and Dumas, 2008), a hydrodynamic model based on the shallow water equations. A total of 5 embedded models with 3 levels of nesting and different

spatial resolutions were selected (Fig. 3a). The second tidal data source was used to cover part of the Atlantic coast of Portugal until the Gulf of Cadiz, which are not included in the tidal atlas. The complement data was taken from the native mesh of the FES2014 model (Carrere et al., 2015) and regridded to 0.004° (Fig. 3b).

In all simulations, the boundary conditions are updated every 3 hours, winds every 1 hour, tidal levels and velocities fields are updated each 30 minutes. The output frequency of the nested model is hourly.

### 105 2.5 Spectral discretization and time steps

The same extended frequency range used in the global grid was employed in the regional mesh to perform all simulations, matching the discretization at the boundary. The extension to higher frequencies is aimed to allow for a better representation of the variability of the wave spectrum for very low wind speeds or very short fetches. At the other end, the purpose of adding lower frequencies is to let the spectrum develop longer wave components for severe storm cases (e.g. Hanafin et al., 2012).

In terms of directional discretization, we used 36 mainly directions (10° resolution), and tests with 24 and 48 directions were employed to verify the effects of the directional resolution.





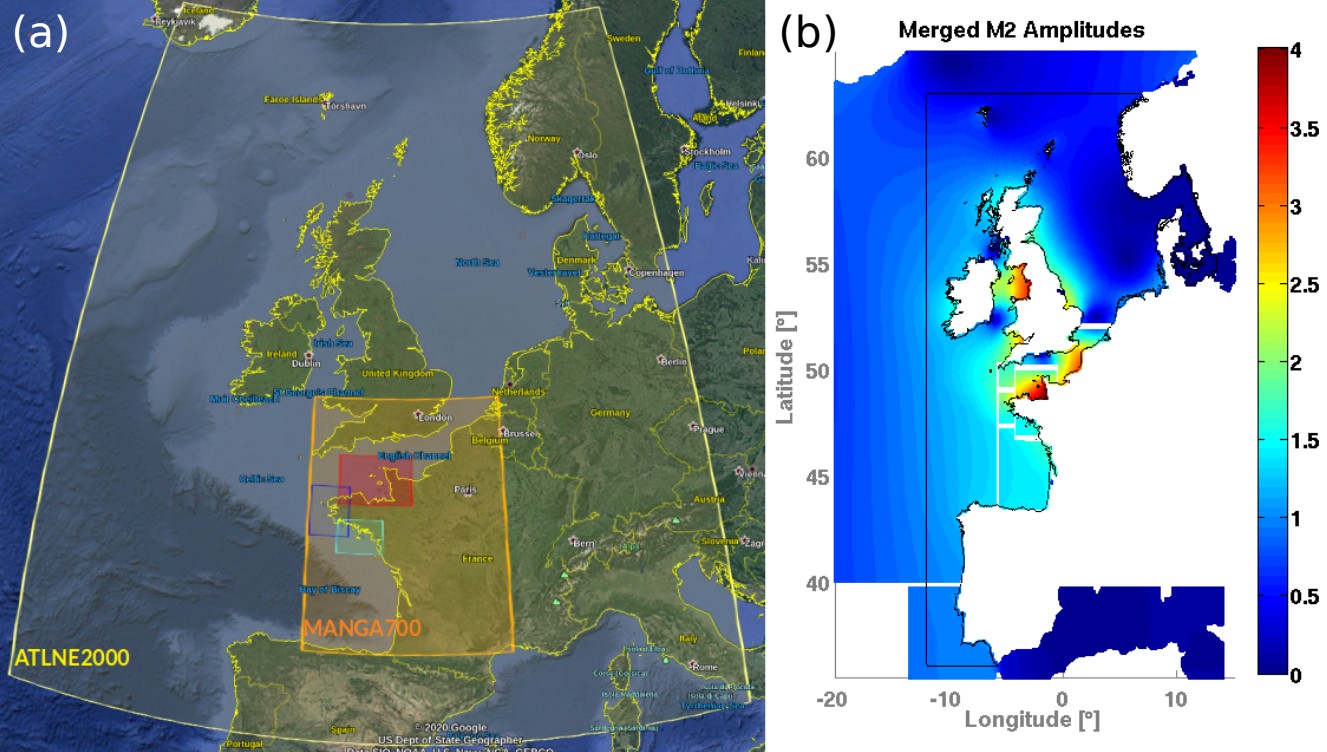

**Figure 3.** (a) Spatial coverage from selected tidal models. Blue, green and red rectangles have a 250 m resolution, the orange and yellow area have resolutions of 700 m and 2000 m respectively. (b)Example of merged tidal harmonics from Ifremer's tidal atlas and FES2014. Map data in (a) are from ©Google Landast / Copernicus. Colorbar in (b) represents M2 amplitude values in meters; black lines show the boundary and coastline polygons.

The source terms are integrated with an adaptative time step that is automatically adjusted in the range 5 to 180 s. We defined the maximum model advection time step to be 30 s, taking into account the minimum mesh triangle area and the presence of strong currents. The refraction time step was set to 15 s. Sensitivity tests with smaller values (not shown) had very limited
impact on the model results.

## 3 Wave data sources

### 3.1 Buoy data

We use 6 French buoys with spectral data provided by CEREMA, and 2 Belgian buoys from which spectra were not available, but besides the usual significant wave height, they provide a low frequency wave height $H_{10}$ (Fig. 4). The $H_{10}$ pararameter
corresponds to a wave height computed for periods from 10 s and longer ($\leq 0.1$ Hz). These sites cover a wide range of depths, current intensities, tidal amplitude levels and proximity to shore, which makes them an appropriate sampling group to evaluate





the overall accuracy of the results (table 1). No assessment of potential instruments' replacements, maintenance periods nor deploy position changes have been taken into account for this study.

To match the frequencies discretization of the spectrum and output frequency (hourly) in WW3, spectral data from the in situ measurements have been first interpolated into the same discrete frequencies used in the model, and then averaged in time to provide hourly output.

**Figure 4.** Buoys location and bathymetry features.(a) Buoys along French coast. (b), (c), (d) and (e) details of French buoys locations. (f) detail of Belgian buoys location. Colorbar shows depths in meters with respect to MSL. Maximum depth on each panel has been selected to enhance bathymetry details.



| Buoy WMO ID | Location name | Longitude [°] | Latitude [°] | Distance to coast [km] | Depth [m] | Data type |
|---|---|---|---|---|---|---|
| 62059 | Cherbourg | -1.6200 | 49.6950 | 4.0 | 28.99 | spectral |
| 62069 | Pierres Noires | -4.96833 | 48.29033 | 15.06 | 67.12 | spectral |
| 62074 | Belle Ile | -3.2850 | 47.2850 | 4.1 | 56.21 | spectral |
| 62078 | P. du Four | -2.7870 | 47.2390 | 19.0 | 37.50 | spectral |
| 62064 | Cap. Ferret | -1.44667 | 44.65250 | 14.7 | 53.45 | spectral |
| 62066 | Anglet | -1.61500 | 43.532166 | 6.7 | 56.77 | spectral |
| – | Westhinder | 2.4358 | 51.381 | 32.3 | 21.90 | $H_{10}$ |
| – | Scheur Wielingen | 3.3022 | 51.401 | 4.75 | 7.80 | $H_{10}$ |

**Table 1.** Spectral buoys ID, location name, position and estimated deploy depth. Distance to coast estimated with respect to continental coast, except for buoy 62074. Deploy depth obtained from model bathymetry interpolated into the buoys' position.

## 3.2 Satellite altimetry data

Given the advantages of altimeters' spatial coverage, the general performance evaluation of the model results was done by comparing results with the ESA Sea State CCI V2 altimeter dataset. We used the "denoised" (Schlembach et al., 2020; Quilfen and Chapron, 2021) significant wave height (SWH) at 1 Hz, to estimate the performance indicators in an along track statistical analysis of the wave heights, and for time averaged values over the complete modelled domain. The adjusted denoised SWH has an along track spatial resolution equivalent to approximately 7 km.





## 4 Model Performance

We use the following statistical parameters: The Root Mean Squared Error (RMSE), Normalized Root Mean Squared Error
(NRMSE), Scatter Index (SI), Mean Bias (BIAS) and the Normalized Mean Bias (NMB):

$$\mathrm{RMSE}(X) \quad = \quad \sqrt{\frac{\sum (X_{\mathrm{mod}} - X_{\mathrm{obs}})^2}{N}} \tag{1}$$

$$\mathrm{NRMSE}(X) \quad = \quad \sqrt{\frac{\sum (X_{\mathrm{mod}} - X_{\mathrm{obs}})^2}{\sum X_{\mathrm{obs}}^2}} \tag{2}$$

$$\mathrm{SI}(X) \quad = \quad \sqrt{\frac{\sum \left[ (X_{\mathrm{mod}} - \overline{X_{\mathrm{mod}}}) - (X_{\mathrm{obs}} - \overline{X_{\mathrm{obs}}}) \right]^2}{\sum X_{\mathrm{obs}}^2}} \tag{3}$$

$$\mathrm{BIAS}(X) \quad = \quad \frac{1}{N} \sum (X_{\mathrm{mod}} - X_{\mathrm{obs}}) \tag{4}$$

$$\mathrm{NMB}(X) \quad = \quad \frac{\sum (X_{\mathrm{mod}} - X_{\mathrm{obs}})}{\sum X_{\mathrm{obs}}} \tag{5}$$

where $X_{\mathrm{obs}}$ are the observed quantities from in situ or satellite measurements, $X_{\mathrm{mod}}$ are the modelled quantities (spectral
values or integrated wave parameters), and N the total amount of analyzed data.

We use the term normalized mean differences (NMD) when using eq. 5 between different model configurations.

### 4.1 Influence of spatial resolution

A comparison between February 2011 mean significant wave heights ($H_s$) fields from the global model described in section
2.4 and our implemented regional model is presented in Fig. 5. To evaluate the differences between models, the output from
the 0.5° global grid was linearly interpolated into the regional mesh nodes before computing the mean $H_s$ and the NMD for
the selected time window.

The most important differences are found on the shelf where complex coastline geometry and bathymetry requires higher
detail to better represent land shadows and wave refraction (NMD in Fig. 5). The largest NMD positive values ($> 20\%$) are
commonly found in the regions sheltered from North Atlantic swells. In the global model, islands and headlands smaller than
the grid size are parameterized as obstructions of the wave energy flux (Chawla and Tolman, 2008). Another direct effect
of using increased spatial resolution can be seen between the Orkney and the Shetland islands. The regional model shows
averaged $H_s$ values of almost 5 m in this area for the analyzed month. On the other hand, the combined effects of the sub grid
obstruction approach and coarse resolution of the global grid, leads to high underestimation of about -20% with respect to our
mesh results.

### 4.2 Adjustments in wind-wave generation and swell dissipation

Alday et al. (2021) adjusted the parameterizations of wind-wave generation and swell dissipation proposed by Ardhuin et al.
(2010) and Leckler et al. (2013), these adjustments were designed to better represent the wave heights measured by altimeters at
global scales. Here we further consider the impact of these modifications on waves in our coastal domain, using five different



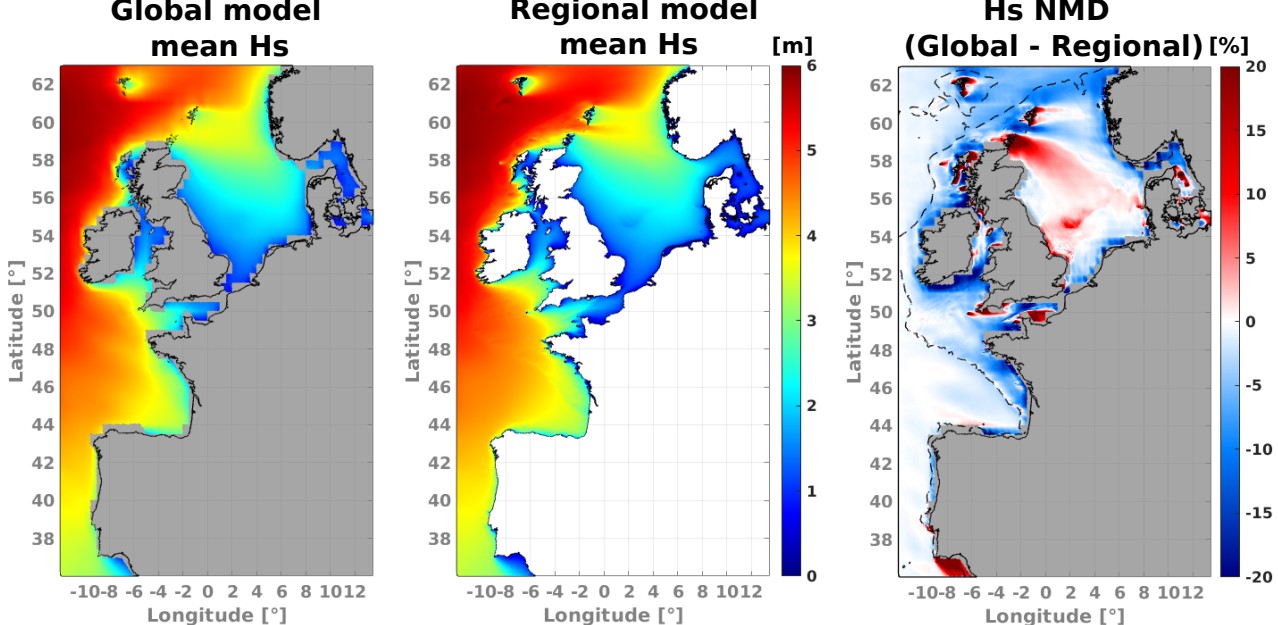

**Figure 5.** Mean $H_s$ fields from global and regional model, and $H_s$ normalized mean differences (Global - Regional). Dashed black lines represent the 400 m depth contours. Areas where no wave data are available from the global grid are highlighted with a gray background in left and right panels. Results for February 2011.

simulations with parameter changes listed in table 2. These changes include an empirical enhancement of the wind speeds above a threshold $U_c$ by the amount $x_c(U_{10} - U_c)$, and a modification of the swell dissipation with a change in the threshold Reynolds number $\mathrm{Re}_c$ that defines the transition from the weak (laminar) to strong (turbulent) swell dissipation and the swell dissipation coefficient $s_7$.

We analyzed model results for two months when extreme sea states have been recorded, February 2011 and January 2014. In February 2011, the extra tropical storm Quirin generated extreme sea states with peak periods exceeding 20 s over the western coasts of Europe. In January 2014, storm Hercules was one of the many storms from a particularly severe winter. This event caused vast coastal damage in the UK (Masselink et al., 2016), and from the Western coast of France to Portugal (Masselink et al., 2015). $H_s$ values exceeded 10 m and peak periods exceeded 20 s (De León and Soares, 2015; Castelle et al., 2015).

Given the characteristics of the selected cases, it is considered that they are suitable to study wave energy fluctuations down to frequencies lower than 0.06 Hz. Although analyses were carried out for February 2011 and January 2014, in this section we only present the results for the later period.

     Despite the similarities between time series of the wave parameters such as $H_s$ and $T_{m02}$ from one test to another, they noticeably differ for extreme values. Yet, the model runs have systematic differences as a function of wave heights or wave

periods, with 5 to 10% deviations for the larger periods and heights that correspond to the most severe storms and associated swells (Fig. 6). In these events, and consistent with the global scale results, the wind enhancement is most effective at correcting





| Test Name | $s_7$ | $Re_c$ | $U_c$ (m s$^{-1}$) | $x_c$ |
|---|---|---|---|---|
| Bm1.75 | $3.60 \times 10^5$ | $1.50 \times 10^5$ | - | - |
| Bm1.75-W02 | $3.60 \times 10^5$ | $1.50 \times 10^5$ | 21 | 1.05 |
| Bm1.75-W03 | $3.60 \times 10^5$ | $1.50 \times 10^5$ | 23 | 1.08 |
| Bm1.75-W04 | $3.60 \times 10^5$ | $1.50 \times 10^5$ | 22 | 1.05 |
| **T475** | $\mathbf{4.32 \times 10^5}$ | $\mathbf{1.15 \times 10^5}$ | 21 | 1.05 |

**Table 2.** Tests for wind correction and swell dissipation parameters, in bold, values leading to T475. All parameters not specified here correspond to the T475 parameter adjustment detailed by Alday et al. (2021). Variables $Re_c$, $U_c$ and $x_c$ correspond to namelist parameters SWELLF7, SWELLF4, WCOR1 and WCOR2 in the WW3 input files (see Appendix A for the full set of parameters). The directional discretization has 24 directions in all of these tests.

the low bias in extreme wave heights and mean periods that is typical of the previous hindcasts. Adjustments to the swell dissipation have a negligible impact when acting only 1000 km or less of propagation within our coastal domain. As shown in Fig. 7, the wind enhancement allows the generation of lower frequency waves. This improves the model accuracy at exposed
buoys 62066, 62074 and 62069, and produces realistic energy levels for frequencies below 0.05 Hz during the extreme events of January 2014. Unfortunately, the correction also produces too much low frequency energy at the shallower buoy 62078. We suspect that dissipative processes in shallow water may be underestimated for these very large periods (Fig. 7e,f).





**Figure 6.** NMB and SI for tests leading to T475 (table 2). Results for January 2014. In (a) and (b) modelled results compared with buoys 62074 and 62069 respectively. $H_s$ bin size is 0.25 m, periods bin size is 0.2 s.





**Figure 7.** Performance parameters for energy levels at each discrete frequency of the spectrum, for tests leading to T475 (table 2). Results for January 2014 at buoys 62066, 62078, 62074 and 62069. In panels (a) to (d) modelled results compared with buoy data. Time series of modelled and measured $H_{20}$ for buoys 62078 in (e) and 62074 in (f).





## 4.3 Wave-Current Interactions

At global scale, the use of ocean surface currents can improve the accuracy of the simulated sea states (Echevarria et al., 2021;
Alday et al., 2021), although a full effect generally requires relatively high spatial resolution that is generally not achievable by
observations and thus models are usually not constrained at the necessary scale (Marechal and Ardhuin, 2020). Adding surface
currents in the simulations can have effects on wave generation due to changes on the relative wind, it can modify the advection
of waves or induce refraction in regions with large current gradients. Given the diverse tidal amplitudes within the modeled
domain, it is expected to have different effect levels over the sea states in different areas. We thus attempt to characterize the
changes of the wave field when tidal currents are taken into account in the simulations. To do so, we look at differences on a
set of wave parameters, namely $H_s$, directional spreading SPR, the peak direction $D_p$ and peak period $T_p$. We first checked
global scale current effects via the boundary conditions, and then focused on tidal current effects within our coastal domain.

To evaluate the effects of global currents on the boundary condition, we analyzed a specific output time with a large Atlantic
swell, and differences between 1 month simulations. The most noticeable changes caused by global currents are obtained for
$H_s$, $D_p$ and directional spreading (Fig. 8 middle panel), with typical differences of the order of 5 %. These differences vanish
when averaged over one month (Fig. 8 right panel).







**Figure 8.** Global currents effects over (a) $H_s$, (b) $D_p$ and (c) directional spreading. In left panel, model output for test using BC generated with global currents from 16 February 2011, 00:00:00 (UTM). NMD results in middle and right panels are for test with BC obtained without global surface currents with respect to test with BC from global grid forced with global currents. Colorbars in middle and left panels represent NMD in [%]. Full simulation duration of tests is 1 month.





The effects of tidal currents within the model domain are generally more important, with some strong local effects caused by the high spatial currents' variability. In contrast to the influence of global currents in the BC, there is a clear increase of the wave fields' differences at each temporal output, that can be larger than +/-10 %. Feature mainly seen along the English

Channel and the Irish Sea (Fig. 9, left panel). Over the entire month, tidal currents induce mean $H_s$ differences of the order of 5 % (Fig. 9, right panel).

The use of tidal currents also proved to have large impact over the peak period ($T_p$), up to 15% differences in Normandy and Liverpool bay, for example, and 8 % mean differences over one month (not shown).

There is a noticeable feature of the wave field along the shelf break, starting at the Bay of Biscay and extending northwards

up to 49°N, which can be seen more clearly through the $D_p$ and $H_s$ fields from Fig. 8b,c (left panel), and particularly by analyzing the effects of tidal currents over the wave heights in Fig. 9c (left panel). The intensities of current used in our model present maximum values of about 0.5 m s$^{-1}$ along the aforementioned area, which is consistent with previously recorded in situ measurements and the expected sharp variation of currents across the shelf break (i.e. Le Cann, 1990). It is thought that the distinct gradients visible in some of the wave parameters are function of the tides' phase and the mean wave direction.

Attempts to identify the presence of this signature with altimeter data is an ongoing subject of study.

Results were further compared against in situ data from January 2014 at buoy 62059 (Fig. 10). Including tidal currents helps to reduce the high energy bias at low frequencies, probably due to an overall reduction of the effective wind input for locally generated waves during the tidal cycle (Fig. 10a). In Fig. 10c is possible to observe the modulation of $H_s$ and $T_{m01}$ caused by the changes in currents intensities and direction (blue line in figure), which in the end helps to reduce the bias of these quantities

compared to the measurements (Fig. 10b). Notice that there is a constant shift in the occurrence of peaks and troughs of $H_s$ and $T_{m01}$ in Fig. 10c. This is thought to be mostly attributed to a slight phase shift in the tidal forcing field, which introduces a slight increase in the RMSE when tidal currents are included in the simulations (not shown).




**Figure 9.** Tidal currents effects over (a) $H_s$, (b) $D_p$ and (c) directional spreading. NMD results obtained with respect to test using tidal currents. In left panel, NMD with respect to model output from 16 February 2011, 00:00:00 (UTM) presented in left panel of Fig. 8. Colorbars represent NMD in [%]. Full simulation duration of tests is 1 month.





**Figure 10.** Evaluation of tidal currents effects on wave energy distribution (a) , $H_s$ and $T_{m01}$ at buoy 62059 (Cherbourg Exterieur). Wave parameters' NMB and time series in (b) and (c) respectively. Results for January 2014. $H_s$ bin size is 0.25 m, $T_{m01}$ bin size is 0.2 s





## 4.4 Effects of spectral directional resolution

The selection of the spectral discretization plays an important role in the characteristics of the simulated sea states obtained
through the integration of the WAE (Tolman, 1995b; Roland and Ardhuin, 2014). Normally, in coastal applications like assessments of wave energy or simulation of storm surges, higher time and spatial variability details are desired, and hence, higher
spatial and spectral resolution is required (e.g. Bertin et al., 2015; Accensi et al., 2021; Wu et al., 2020). Nevertheless, the
quality of the results may be affected by the characteristics of the used BC.

We analyzed the changes in the energy distribution of the directional spectrum and the wave field evolution due to different
directional resolution values in our mesh and in the BC. The different BC tests are aimed to identify potential effects when
coarser resolution is used at global scale, and then interpolation is applied to match the resolution of the nested mesh (this is
done in WW3). Then, to eliminate the potential influence of energy interpolation at the boundary, we verified the effects on
wave propagation within the mesh domain keeping consistent resolutions at the BC and the nested model. Tests' specifications
are defined in table 3. Variations in the energy distribution due to lower resolution in the BC are presented in Fig. 11, comparing

| Test Name | Number of directions | Directional resolution [°] | Number of directions in BC | Directional resolution in BC [°] |
|:---:|:---:|:---:|:---:|:---:|
| 24D24BC | 24 | 15 | 24 | 15 |
| 36D24BC | 36 | 10 | 24 | 15 |
| 36D36BC | 36 | 10 | 36 | 10 |
| 48D24BC | 48 | 7.5 | 24 | 15 |
| 48D48BC | 48 | 7.5 | 48 | 7.5 |

**Table 3.** Tests for spectral directional resolution effects. All parameters not specified here correspond to test T475. When directional resolution of the boundary conditions (BC) is lower than in the mesh, interpolation is applied at the boundary to match the resolution of the nested model.

BC with 24 spectral directions with respect to 36. A set of 4 locations were selected: At the boundary (named node W12N56),
and along the French coast nodes 62074 (Belle Ile), 62069 (Pierres Noires) and 62059 (Cherbourg). Bathymetry details of these
locations presented in Fig. 4. Here we present results for January 2014, but the analysis was also conducted with simulations
for February 2011.

At the boundary, most of the NMD of energy traveling outside the domain are related to very low levels of spectral energy
(angles $> 270°$ and $< 360°$, Fig. 11a right panels). This has negligible effects over of the already analyzed wave parameters
(e.g. $H_s$, $D_m$, SPR). For waves traveling into the domain , only high NMD ($> +/- 10$ %) are observed at lower frequencies
($< 0.1$ Hz) between directions $20°$ and $150°$ (Fig. 11a right panel), which corresponds to the area with higher mean energy at





this location for January 2014 (defined by the contours in Mean Energy panel of Fig. 11a). We found that this effect is still present in nearshore areas exposed to the incoming swells from the North Atlantic (nodes 62074 and 62069), although with
an overall narrower directional range attributed mostly to the bathymetry induced refraction that tends to "align" the arriving waves (Fig. 11b,c).

No significant changes in energy distribution were found at node 62059, for each output time and for the full simulation NMD (Fig. 11d). This is expected since at Cherbourg the sea state characteristics are mostly driven by the local winds.

To further assess potential changes introduced in wave parameters, we analysed the differences in fields of $H_s$, $T_p$, SPR, $D_p$,
and the mean direction $D_m$ (not shown; Fig. 12). Using coarser directional resolution in the BC has minor effects over wave parameters integrated along the complete frequency range (e.g. $D_m$ or $H_s$; Fig. 12b, top panel). Differences in the results are exacerbated when BC with 24 directions are interpolated into 48 (right panels in Fig. 12a,b) but in general NMD and NRMSD between tests remained below $+/-2.5$ %, with the exception of $T_p$ that presented the largest NRMSD.

Even though the magnitude of these quantities remain fairly consistent, interpolating BC with coarser directional resolution
affects the characteristics of the wave fields propagating into the domain. This is attributed to slight changes in the wave celerity (C=$gT/2\pi$ in deep waters) due to frequency shifts in the neighborhood of the energy peak (Fig. 11a,b,c, Energy difference panels).







**Figure 11.** Effects of boundary conditions with lower directional resolution at different output locations. (a) Boundary node W12N56 (Lon.: 12°, Lat.: 56°) (b) 62074 (Belle Ile), (c) 62069 (Pierres Noires), (d) 62059 (Cherbourg). Differences and NMD (36D24BC-36D36BC) computed for January 2014. White contours marking energy levels on left panels are the same plotted in black on the corresponding right panels for energy difference and NMD. Direction convention is towards energy is traveling to.




**Figure 12.** (a) Normalized mean differences (NMD) and (b) normalized root mean squared differences (NRMSD) between tests 36D24BC-36D36BC and 48D24BC-48D48BC. Analyzed period : February 2011. Colorbars represent changes in quantities between tests in [%] units.



The analysis of directional resolution of the mesh is mainly focused on the effects of the Garden Sprinkler Effect (GSE) on wave propagation. This phenomenon is observed as a separation or disintegration of continuous swell fields propagated with a

discretized spectral wave model (Booij and Holthuijsen, 1987; Tolman, 2002). The generation of the GSE is namely linked to the spectral resolution and the selected numerical scheme to solve the WAE. Currently there are no GSE alleviation methods available for unstructured grids in WW3.

A good example was found during February 1st 2011, where a strong swell from the North Atlantic arrived to the northern coast of Scotland. In Fig. 13a we present an instant (13:00:00 UTC) of the event using 3 different discretizations from tests

24D24BC, 36D36BC, 48D48BC (table 3). The GSE can be observed to the East of the Orkney and Shetland Islands towards the Norwegian Sea (between latitudes 58° and 61°) when 24 directions are employed (Fig. 13a, left panel).

The impact of the GSE was assessed by comparing the results against the output from a model with higher directional resolution. Via a straight forward difference between tests, is possible to visualize changes of the $H_s$ field caused by the spurious wave propagation pattern (Fig. 13b). Comparing tests 24D24BC with 36D36BC, and for this particular scenario,

differences in wave height can reach +/-0.2 m (roughly +/-5%) as the swell approaches Norway, between longitudes 2° to 4° (Fig. 13b, left panel). These values are only slightly higher when comparing tests with 24 to 48 directions (Fig. 13b, middle panel). Between 36D36BC and 48D8BC, only minor $H_s$ changes are generated (< 0.05 m; Fig. 13b, right panel).

The mild repercussion of the GSE over the $H_s$ field in the present results shouldn't be generalized, since this phenomenon could be intensified depending on the incoming swell conditions. Our findings suggest that using a directional resolution of

10° is enough to mitigate the effects of the GSE. It is relevant to point out that, for example, the required computation time in 36D36BC is 40% higher than in 24D24BC, a considerably elevated cost for potential operational (forecasting) applications.





**Figure 13.** (a) $H_s$ field from 1 February 2011, 13:00:00 (UTM) for different directional resolution tests specified in table 3. (b) Differences in $H_s$ fields presented in (a). Offshore swell conditions (to west of the Orkney and Shetland Islands): $T_p = \sim 14$ s, $D_m = \sim 260°$.



## 4.5 Bottom friction effects

Over the continental shelf, in intermediate to shallow waters, the evolution of the wave fields becomes influenced by the bottom characteristics. In the absence of strong wind seas and outside the surf zone, dissipation of energy is mainly induced by bottom
roughness effects. We thus try to quantify the effects of including the bottom friction sink term in the WAE.

To provide a general view, we compared model output from 1 year tests with the 1 Hz altimeter data from the ESA Sea State CCI V2 datset. For this particular analysis 1 year simulations were required in order to have at least a minimum of 5 satellite measurements to compare with the regridded WW3 $H_s$ fields at 0.1°. Only altimeter measurements at least 10 km away from the coastline were considered to avoid potential data with high noise to signal ratio.

Bottom friction effects were included through the SHOWEX parameterization proposed in Ardhuin et al. (2003). This expression was initially developed for sandy bottoms based on the eddy viscosity model of Grant and Madsen (1979) and includes a decomposed roughness parameterization for ripple formation and sheet flow. In WW3 it has been implemented including a sub grid parameterization for water depth variability following Tolman (1995a). The bottom friction source term can be written as follows:

$$S_{\mathrm{bot}} = \quad f_e u_{\mathrm{b,rms}} \frac{\sigma^2}{2g \times \sinh^2(kd)} N(k,\theta) \tag{6}$$

with,

$$f_e = \quad \frac{\kappa^2}{2\left(\mathrm{Ker}^2(2\sqrt{z_o/l}) + \mathrm{Kei}^2(2\sqrt{z_o/l})\right)} \tag{7}$$

and

$$z_o/l = \quad \sqrt{\frac{2}{f_e}} \frac{k_N}{30\kappa a_{\mathrm{b,rms}}} \tag{8}$$

When the Shields number $\psi$ is $>= A_3\psi_c$, the Nikuradse roughness $k_N$ is taken as the sum of the ripple roughness $k_r$ and a sheet flow roughness $k_s$:

$$k_r = \quad a_{\mathrm{b,rms}} \times A_1 \left(\psi/\psi_c\right)^{A_2} \tag{9}$$

$$k_s = \quad 0.57 \frac{u_{\mathrm{b,rms}}^{2.8} a_{\mathrm{b,rms}}^{-0.4}}{[g(s-1)]^{1.4}(2\pi)^2} \tag{10}$$

where,

$$\psi = \quad f_w u_{\mathrm{b,rms}}^2 /[g(s-1)D_{50}] \tag{11}$$

$$\psi_c = \quad \frac{0.3}{1+1.2D_*} + 0.55[1 - \exp(-0.02D_*)] \tag{12}$$





with,

$$D_* = \quad D_{50} \left[ \frac{g(s-1)}{\nu^2} \right]^{1/3} \qquad (13)$$

In table 4 we present a set of empirical parameters originally taken from Ardhuin et al. (2003) where we have particularly modified $A_5$ to 0.04. $D_{50}$ is the median sediment size in meters defined at each node of the unstructured grid (see Fig. 2). A full description of the terms in eq. 6 to 13 can be found in Ardhuin et al. (2003) and in the WW3 user manual (The WAVEWATCH III $^{\circledR}$ Development Group, 2019).

| Parameter | WW3 variable | value |
|:---------:|:------------:|:-----:|
| $A_1$ | RIPFAC1 | 0.4 |
| $A_3$ | RIPFAC3 | 1.2 |
| $A_4$ | RIPFAC4 | 0.05 |
| $A_5$ | BOTROUGHMIN | 0.04 |
| $A_6$ | BOTROUGHFAC | 1.00 |

**Table 4.** List of empirical parameters used in SHOWEX bottom friction parameterization. The WW3 variables' names correspond to the keyword used in the model's BT4 namelist.

To assess the effects of the bottom friction parameterization, we first compared 1 year simulations with and without dissipation to verify changes in the wave field. In Fig. 14a we present the $H_s$ mean bias obtained by comparing with Saral (year 2014) for the full domain. A clear reduction of the wave heights bias is detected in the south of the North Sea. In this area, we found that $H_s$ mean differences between results with and without bottom friction can be of 0.3 m and higher. Analysis with other alimeters (e.g. Jason-2 and Envisat) for year 2011 show consistent results.

In general, with altimeter data most relevant changes in wave heights, when bottom friction is included, are detected for depths smaller than 50 m. We found a couple of Envisat tracks passing almost parallel off the coast of La Rochelle and close to Ile de Yeu (Fig. 14b). In both locations the use of the bottom friction parameterization, with the defined $D_{50}$, helps to reduce the $H_s$ mean bias. These results are consistent with the findings of Roland and Ardhuin (2014) for this area based on buoy data.

We picked 3 locations to compare our results with in situ measurements, buoy 62078 on the Altantic French coast, and buoys Westhinder and Scheur Weilingen deployed in shallower depths along the coast of Belgium (Fig. 4).

For buoy 62068 we first compared the full $H_s$ time series of in situ data against simulations with and without bottom friction effects. Reductions in the wave height's NMB and NRMSE of respectively 4.5 % and 5.0% are obtained when bottom friction and the sediment size map are included (Fig. 15a,c). Nevertheless, most significant changes in the modeled $H_s$ appear at wave heights roughly larger than 3 m. We then selected an ad hoc $H_s$ threshold of 3.5 m to define "extreme" sea states and analyze the effects of the parameterization over the events on which dissipation due to wave-bottom interactions is dominant. For these events, a wave height bias and RMSE reduction of about 0.3 m, with a decrease of about 8% and 5.3% in the NMB and SI is obtained when the SHOWEX dissipation term is used (Fig. 15b). Moreover, we found good agreement between the occurrences





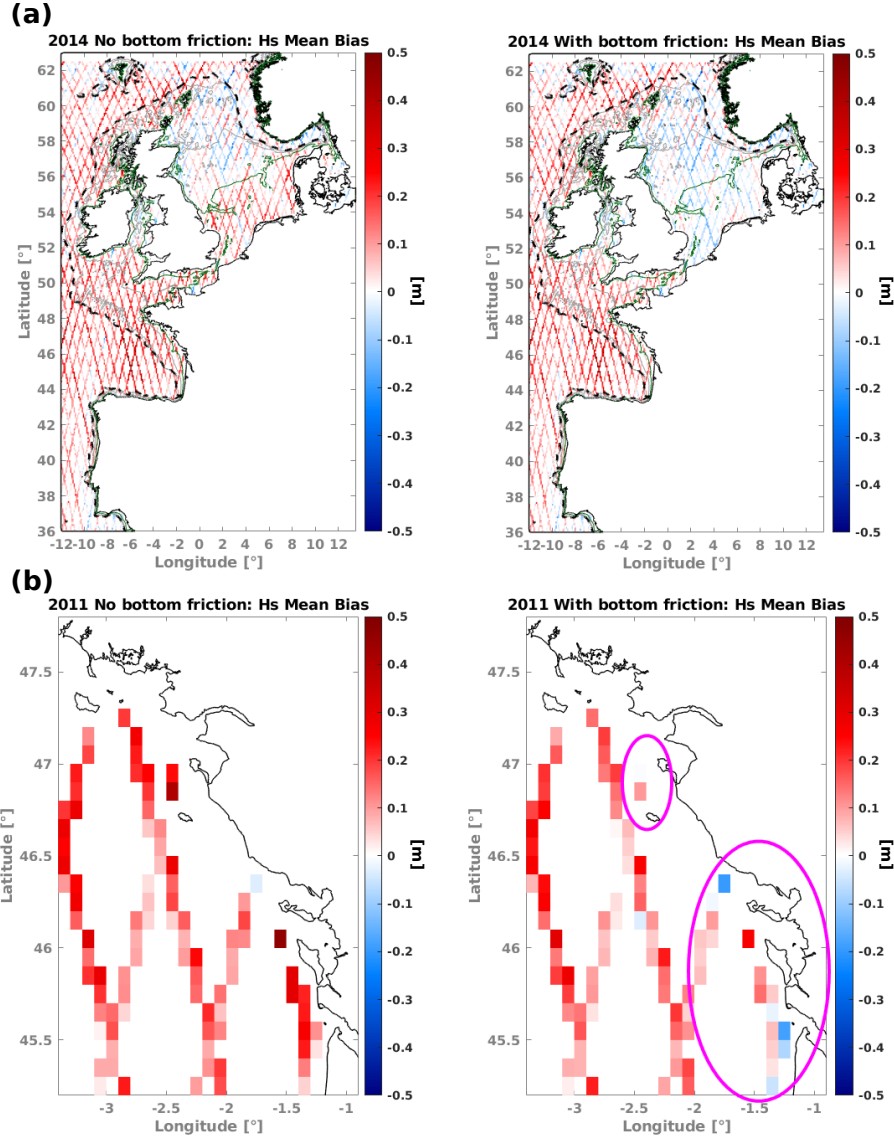

**Figure 14.** $H_s$ bias (WW3 - altimeter) computed with (a) Saral year 2014 and (b) Envisat year 2011. Dashed black lines show 200 m depth contour, green lines the 50 m depth contours, and gray lines depth contours from 100 to 150 m depth. Magenta ovals in (b) highlight areas with mayor bias reduction.

of the Shields number $\psi$ exceeding its critical value $\psi_c$ (Fig. 15d) and the occurrences of extreme sea states with $H_s > 3.5$ m

(Fig. 15c) especially between January and March 2014. Notice how $f_e u_{b,rms}$ increases every time the critical Shields number is exceeded, giving more weight to the $S_{bot}$ term in the WAE. In this case, the definition of extreme events helps to identify when





the effects of bottom friction becomes relevant, since larger $H_s$ are normally related to longer wave lengths, thus wave-bottom interactions start at deeper depths.

At Westhinder and Scheur Weilingen we analyzed the dissipation effects over components of the spectrum with periods
longer than 10 s comparing $H_{10}$ values. For these locations we also compare with simulations using the JONSWAP bottom friction parameterization (Hasselmann et al., 1973; Tolman, 1991) with its default values (Fig.16). Wave energy for components longer than 10 s is clearly over estimated when no bottom friction is taken into account. The effect is visible at both analyzed depths. At Westhinder both parameterizations have similar effects, but at the shallower buoy location (Scheur Weilingen) the use of SHOWEX and the selected $D_{50}$ introduce a negative bias of $H_{10} > 0.5$ m which could be related to an overestimation
of the sediment mean size in this area.





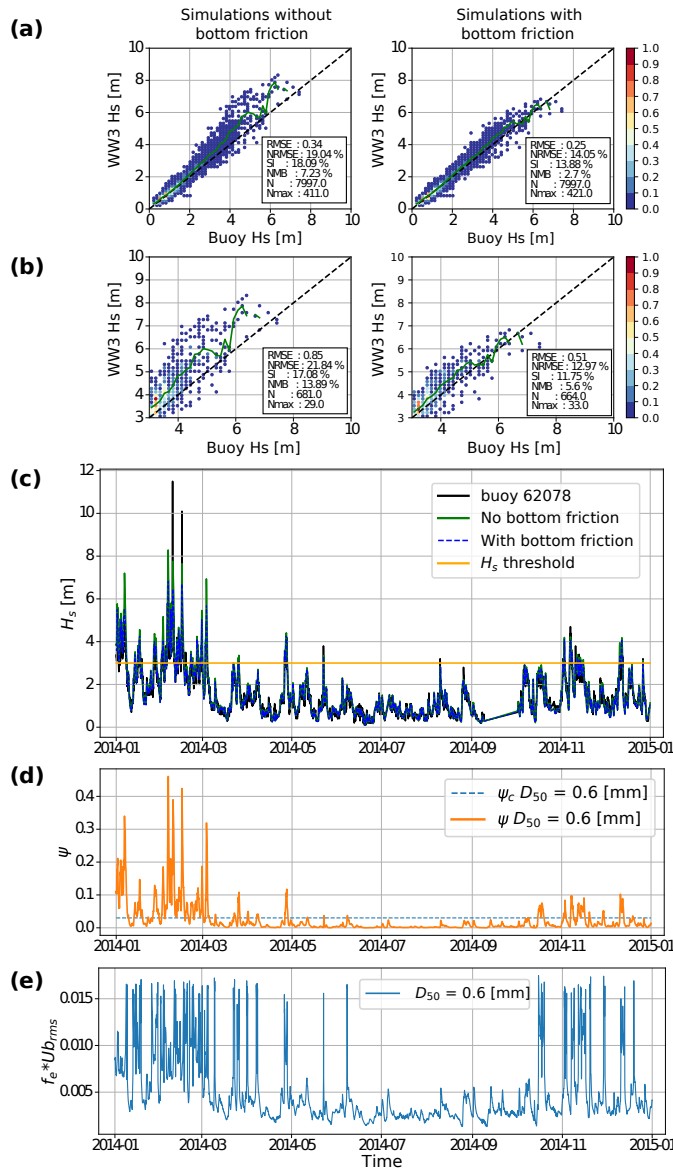

**Figure 15.** Bottom friction effects at buoy 62078 (year 2014). Performance analysis using (a) complete time series and (b) extreme events ($H_s > 3.5$ m). (c) $H_s$ time series for cases with and without SHOWEX parameterization. Time series of (d) Shields number $\psi$ and (e) dissipation term $f_e u_{b,\mathrm{rms}}$. In (a) and (b) green line shows the modelled averaged values at each 0.15 m wave height bin. Colorbars represent the wave heights frequency of occurrence normalized by the total amount of analyzed data N. Time series in (d) and (e) computed with WW3's frequency spectrum following eq. 6 to 13. $D_{50}$ taken from bottom sediment map (Fig. 2). Blue dashed line in (d) represents the critical Shields number.

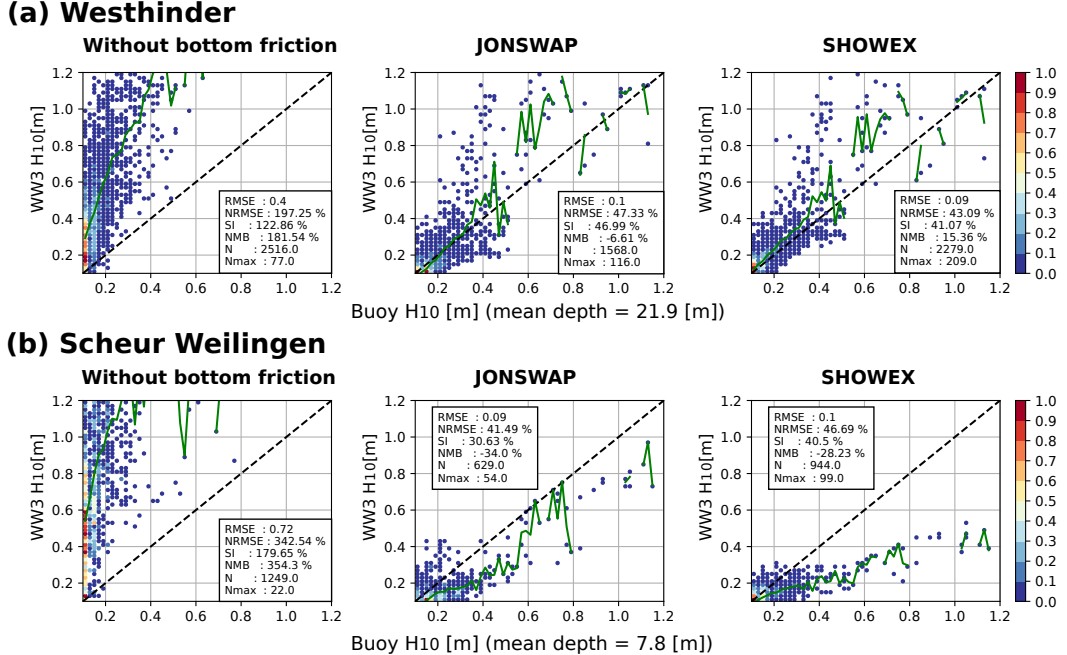

**Figure 16.** WW3-Buoy $H_{10}$ comparison for tests without bottom friction, using default JONSWAP and with SHOWEX parameterization including the implemented bottom sediment map. Results for (a) Westhinder and (b) Scheur Weilingen buoy location for year 2014. Green line shows the modelled averaged values at each 0.02 m wave height bin. Colorbars represent the wave heights occurrences normalized by the total amount of analyzed data N.



## 5   Model validation with altimetry data

Satellite altimetry provides a unique resource of worldwide wave heights' measurements. The integration and inter calibration of past and ongoing missions have allowed to continuously extend the coverage of measured data in space and time (Ribal and Young, 2019; Dodet et al., 2020). These datasets have been commonly used in open ocean applications to improve our

understanding of the sea states globally. On the other hand, their application in coastal (especially nearshore) areas has been very limited due to increased noise levels in the return signal. What is considered as noise is actually the detection of the non Gaussian land surface, which makes it difficult to retrieve the waves geophysical signal in the radar footprint.

The Sea State CCI V2 dataset employs the WHALES partial waveform retracking algorithm, more effective for reducing the intrinsic noise of the return signal, and suitable for coastal applications (Schlembach et al., 2020; Passaro et al., 2021). The

vast amount of measurements available at distances from the coast lines down to 5 km and less implies also a large coverage of measured wave heights in shallower depth areas, providing a broader description than local in situ records. Making use of the coverage and improvements in this altimeter product, we analyzed the performance of our mesh over part of the wave hindcast described in Accensi et al. (2021).

We analyzed 3 zones of the modeled area: Bay of Biscay, North Sea and English Channel. The purpose of the defined

zones is to assess the performance of the model in different wave generation and propagation conditions. The Bay of Biscay is constantly exposed to swells radiated from the North Atlantic. At the North Sea, wave conditions are dominated by the local winds blowing over a well defined fetch and partially influenced by the swells from the Norwegian Sea. Finally, at the English Channel, most of swells' energy arriving from the North Atlantic is blocked, refracted and dissipated on its western end, local waves are generated over a very short fetch, and it is highly influenced by its tidal regime.

Using an along track comparison of the modeled $H_s$ with respect to the altimetry derived SWH, the NMB and SI were computed per altimeter mission as function of the distance to the coast, using bins of 1 km and considering SWH > 1 m. To provide an idea of the lower and upper bound values of NMB and SI from distances of 1 km offshore up to 80 km, the performance parameters were computed over the complete available years of data per mission until 2018: from 2002 to 2012 for Jason-1 and Envisat, 2008 to 2017 for Jason-2, 2013 to 2018 for Saral, and from 2016 to 2018 for Jason-3 (Fig. 17).

From distances to the coast of 15 km and more we noticed a constant positive bias ranging from 2 to 6% in the Bay of Biscay, and in some cases going up to ~8% in the English Channel. At the North Sea bias changes are more constrained between +/-2% (Fig. 17a). The positive bias in the Bay of Biscay is thought to be related to the BC obtained from the global hindcast using T475, which was calibrated with the Jason-2 data from CCI V1. This data was indeed found to overestimate SWH recorded by offshore buoy measurements (Dodet et al., 2020), which has been corrected in V2. The English Channel stands out as

high NMB and SI area which is thought to be caused by the reduced amount of valid altimeter measurements in this area and inaccuracies of the forcing fields. Finally, less influenced by the BC and with an extended fetch for wave growth, the North Sea presents the lowest NMB values, which along with the more constrained SI (Fig. 17b) shows the good performance of the proposed parameterization and model setup in this area.





An overall NMB decrease is observed for distances to the coast smaller than 15 km, which implies that in general the
altimeters' SWH are higher than the modeled $H_s$. Differences that are particularly more accentuated at the Bay of Biscay
at offshore distances $< 10$ km. Even with the higher uncertainty of modeled/measured wave heights closer to the shore, the
available altimetry data down to ∼6 km offshore still provides unprecedented access to coastal information that, even at this
early stage, allows to evaluate the model performance.

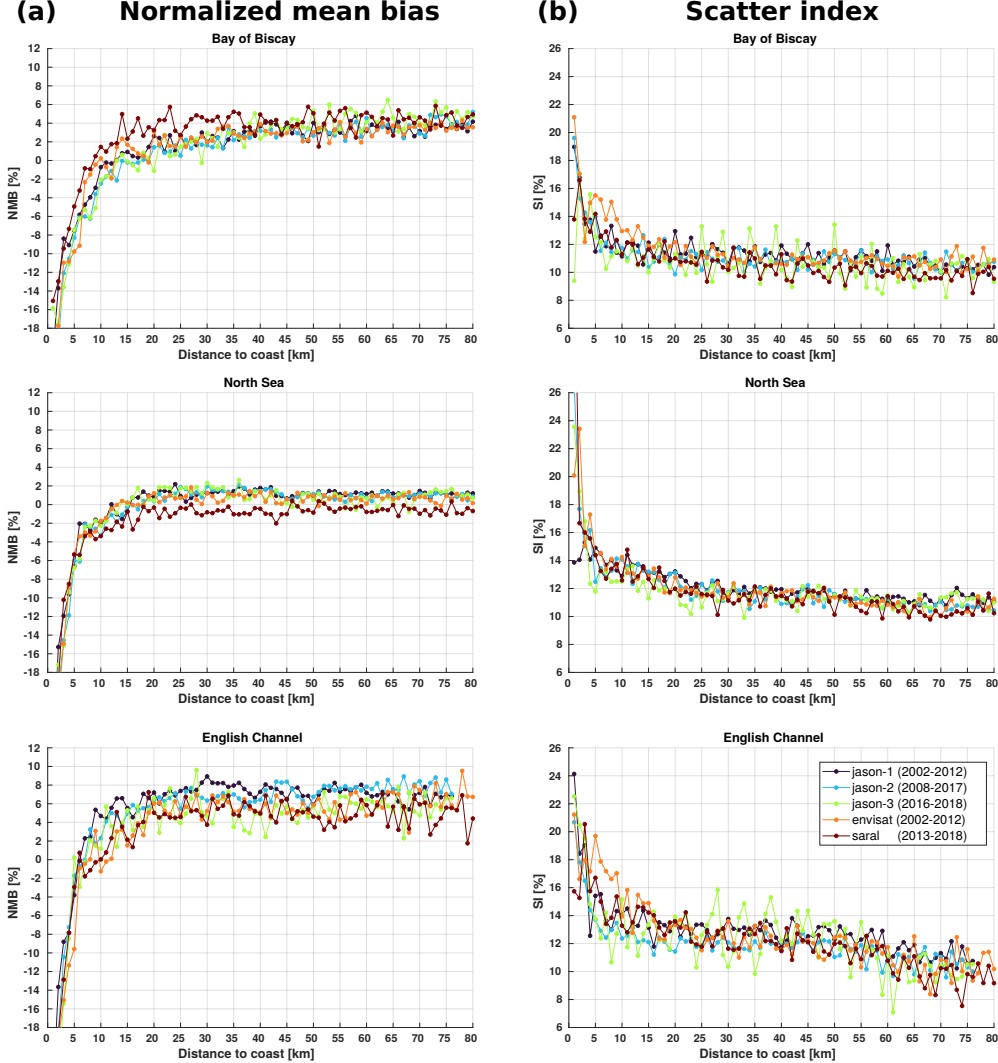

**Figure 17.** (a) NMB and (b) SI of $H_s$ as function of distance to the coast (WW3 - altimeter SWH). Bins' width is 1 km





## 6    Conclusions

In the present study we investigated the drivers of model errors in coastal areas, and how choices of parameterization, forcing, spectral and spatial resolution, and boundary conditions affect the characteristics of the simulated sea states. Extensive sensitivity analyses were carried out with a high resolution regional wave model for European coastal waters using the WAVEWATCH III framework. The performed tests and analyses were aimed to assess when and where the choices in the model setup have a significant effect in regions where wave interactions with complex bathymetry, tidal currents and bottom roughness become

important in wave propagation.

Overall, spatial resolution is one of the most important elements in shallow depth areas. We found that higher spatial resolution adequate to solve bathymetry features and explicitly solving coastlines can introduce changes in $H_s$ estimations of about 20% when compared to lower resolution models. Differences become more significant below 400 m depth, in areas where refraction and diffraction are dominant, or in regions sheltered from the most frequent swell conditions.

Changes in the energy distribution of the spectrum were analyzed mainly from two points of view, introduced by modifications in the parameterization, and due to changes in directional resolution. Modification of the swell dissipation terms did not impact significantly the wave energy distribution in the regional domain, although their effect become important at global scales (Alday et al., 2021). In general, the applied enhancement to intensities higher than 21 m s$^{-1}$ in the ERA5 wind fields improves the model accuracy at swell-exposed locations, helping to reproduce realistic energy levels for frequencies lower than

0.05 Hz, partially solving their otherwise high under estimation (more than -50% in some cases). These findings suggest that the considerations taken to generate the boundary conditions at global scale, are one of the most important factors on shorelines exposed to waves from the North Atlantic.

Spectral energy differences due to directional resolution choices are larger than 10% at frequencies lower than 0.1 Hz. Effect that is visible from the boundary to the nearshore in zones influenced by the BC. Differences in wave parameters (SPR, $T_p$,

$D_p$) observed between model tests suggest that the proper selection of directions to define the BC and within the nested model will help to reduce random errors. It was also found that with 10°resolution, the GSE is successfully alleviated in the mesh.

Within areas with large tidal amplitudes, including tidal forcing (currents and levels) typically change wave parameters by about 10% at each output time, and locally much more (e.g. Ardhuin et al., 2012). These differences are reduced for $H_s$ and $D_p$ for a monthly average, but can still be larger than 5% for the SPR and $T_p$. These findings imply that even if the average

wave heights might be well estimated without tidal forcing, the propagation and evolution of the wave fields will be different. This can be observed in the $H_s$ and $T_{m01}$ time series at buoy 62059 (Fig. 10).

Comparing with wave heights retrieved from altimeter data with 1 year simulations, we identified areas influenced by bottom friction dissipation by looking at changes in $H_s$. We found that these changes can be observed at depths smaller than 50 m. In shallower areas of the North Sea and some sections of the Atlantic coast of France, including the SHOWEX bottom friction

parameterization helps to reduce the $H_s$ bias. Comparisons between model and in situ measurements of $H_10$ revealed an underestimation of the wave energy in the low frequency bands in very shallow areas. This effect could be related to a higher



sensitivity of the SHOWEX parameterization in very shallow depths, thus, dissipation induced in longer wave components is over estimated with our current model setup.

Using 5 available missions from the Sea State CCI V2 dataset we performed a validation of the modelled wave height as function of the distance to the coast, between years 2002 to 2018. We observed an overall increase of $H_s$ differences with our model for distances to the coast smaller than 10 km that can reach -8% (in average) at 5 km from the coast. These differences are likely due to increased uncertainties in altimeter measurements within the last 10km from the coast, where coastal features are known to strongly impact radar waveforms (Vignudelli et al., 2019).

We found that in many cases time averaged differences between model setups or with respect to in situ data are small, but these differences can be significant at each output time, implying that the time evolution of the sea states is in fact different. This could partially explain cases with low bias and still larger random errors (e.g. SI) in some locations, when modelled wave parameters are compared with measurements.

Due to the different characteristics of the modelled domain (e.g. bathymetry features, bottom sediment type, fetch and tidal amplitudes) the factors driving the accuracy of the model cannot be completely generalized. Instead, through the proposed analyses we have identified where changes in the wave field characteristics are more significant with different choices in forcing, resolution and parameterizations. Yet, it is not straightforward to assess how the combination of these choices can potentially compensate errors in the simulations. We find that boundary condition effects are most easily evaluated at deep water or partially sheltered locations (see also Crosby et al., 2017), while separating bottom friction from other effects will require a further analysis of specific swell events.

*Data availability.* The used coast line polygons, bathymetric data, bottom sediment type maps and buoy data have been take from the following web portals:

– OpenStreetMap coast line polygons:

https://osmdata.openstreetmap.de/data/coastlines.html

– EMODnet terrain model: https://portal.emodnet-bathymetry.eu

– HOMONIM bathymetric data:

https://diffusion.shom.fr/pro/risquesbathymetrie/mnt-facade-atl-homonim.html

– EMODnet bottom sediment:

https://www.emodnet-geology.eu/data-products/seabed-substrates/

– Buoys with spectral data provided by CMEMS In Situ TAC:

http://www.marineinsitu.eu/dashboard/

– Long period wave heights data from the Agency of Maritime and Coastal Services (Agentschap Maritieme Dienstverlening en Kust):

https://meetnetvlaamsebanken.be/





*Author contributions.* M. Alday and F. Ardhuin wrote the manuscript draft and analyzed the model results; G. Dodet provided the theoretical and technical background of the used altimeter product; M. Alday and G. Dodet analyzed the altimeter data; M. Alday and M. Accensi implemented the high-resolution coastal model. M. Alday, F. Ardhuin, G. Dodet and M. Accensi reviewed and edited the manuscript.


*Competing interests.* The authors declare that they have no conflict of interest.

*Acknowledgements.* The authors would like to thank F. Lyard (LEGOS) who provided the native mesh of FES2014 global tide model. We also thank Aron Roland for his insights on the manuscript and the BGS IT&E team for their support during the mesh construction.

## Appendix A: Detailed model implementation

All simulations' results presented were generated using the unstructured grid WAVEWATCH III model version 7.0. The following compilation switches were included:

- Physical parameterizations: LN1 ST4 STAB0 NL1 BT4 DB1 MLIM TR0 BS0 REF1 WCOR RWND TIDE

- Advection scheme: UQ

- Numerical choices: F90 NOGRB NC4 SCRIP SCRIPNC SHRD TRKNC O0 O1 O2 O2a O2b O2c O3 O4 O5 O6 O7

In our tests, we used a few different combinations of the swell dissipation terms SWELLF7 and SWELLF4 of the ST4 parameterization (section 4.2). Here we present the model namelist with its final values as defined in T475:

- Wave growth and swell dissipation (SIN4 namelist): BETAMAX = 1.75, SWELLF = 0.66, TAUWSHELTER = 0.3, SWELLF3 = 0.022, SWELLF4 = 115000.0, SWELLF7 = 432000.00

- Wave reflexion parameters (REF1 namelist): REFCOAST = 0.05, REFCOSPSTRAIGHT = 4, REFFREQ = 1.0, REFMAP
= 0.0, REFSLOPE = 0.03, REFSUBGRID = 0.1, REFRMAX = 0.5

- SHOWEX parameterization (SBT4 namelist): SEDMAPD50 = T, BOTROUGHMIN = 0.0400, BOTROUGHFAC = 1.0

- Unstructured grid options (UNST namelist): UGBCCFL = F, UGOBCAUTO = T, UGOBCDEPTH = -15.0, EXPFSN = T

- Wind correction and others (MISC namelist): NOSW = 6, WCOR1= 21., WCOR2=1.05



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
