# Peer review of "Accuracy of numerical wave model results: Application to the Atlantic coasts of Europe"

_EGUsphere, 2022_

## Author Comment (AC1)

**COMMENTS FROM REVIEWER #1:**

Review of "Accuracy of numerical wave model results: Application to the Atlantic coasts of Europe"a paper by Alday and 3 co-authors.

**Summary:**

The authors design an unstructured grid implementation of WW3 for the European coastline. They evaluate the model against buoys and altimetry. They perform sensitivity analyses to determine impact of various things. Topics studied include: simple wind adjustments, swell dissipation settings, tidal currents effects, directional resolution, and bottom friction sensitivity.

**Recommendation:**

I'm not very familiar with the expectations of the journal, especially re: what types of papers are OK. I have a generally positive impression of the paper. So, I'd like to give the authors the benefit of the doubt, and recommend "accept with minor revisions"

**General comments:**

Some of the comparisons seem rather workaday, like the sort of thing that would go into a tech report or dissertation. But it's OK, I think. And they are not just dry comparisons: the authors put substantial thought into them, so nothing comes across as especially unnecessary.

The literature review is sufficient.

The comparisons are careful, the discussion is straightforward/honest, and the authors don't over-sell the outcomes.

The findings are potentially useful for people doing similar modeling.

The use of English is very good. There are a few awkward or broken sentences (e.g. lines 310, 393-394) and minor style errors (e.g. lines 21, 27, 409, 412), but not more than ~ 1 problem per page.

The purpose of the study meanders.

There are a lot of evaluations, but they are a grab-bag that are only similar insofar as they are for the same system. I put them in four broad categories:

1) differences vs. global model (geographic resolution)

2) impact of design choices, where choice is not clear a priori (swell dissipation setting, directional resolution, BC formatting, wind adjustments)
3) impact of including/excluding effect, where it is more or less accepted a priori that the effect should be included (bottom friction, tidal current)

4) general evaluation of the model in its final form (altimetry comparison).

I don't have a good suggestion for how to address this. I suppose the authors do not want to stick to a consistent type of content, since they would lose half the study.

I really like the evaluation of the model against altimetry, sorted by distance from the coast. Excellent.

OBS.1. There are a number of unexplained acronyms/initialisms. Perhaps a glossary would work better than rigorous in-line explanation. I leave this to the editor to decide.

**R1: We have reduced significantly the use of acronyms in order to improve readability, for example**

- **the number of occurrences of "NMD" (normalized mean difference) has been reduced from 16 to 5**
- **the 6 occurrences of WAE (Wave Action Equation) have been removed,**
- **The 3 occurrences of MSL have been removed**
- **10 occurrences of NMB have been removed**
- **10 occurrences of SI have been removed**
- **20 occurrences of SWH or H_s have been removed**

OBS.2. When printed out, some of the figures are hard to see, like Fig 7 a-d, where lines are hard to distinguish.

**R2: Some figures were indeed a bit busy. We have chosen to simplify the figures to make it possible to magnify them. In the case of Figure 7 the normalized bias plots were removed.**

**Specific comments.**

OBS.3. 52: "CFL...minimum time step". Wouldn't CFL dictate the maximum time step?

**R3: This sentence was rewritten for clarity,**

*This coarsening allowed a lower Courant-Friedrich-Lewy (CFL) number, which makes it possible to use larger time step for wave propagation, 13~s in this case for our lowest frequencies, but it also implies that details of the Norwegian coastline are not as well resolved.*

**Indeed the actual time step used at run time is automatically adjusted between a maximum value and whatever it takes to keep the CFL number under 1, and this adjustment is done separately for each spectral component (higher frequencies that propagate slower are typically integrated with higher time steps).**

OBS.4. Same paragraph. Why is the implicit scheme not used? It should be mentioned, at least.

**R4: We have added the following paragraph after Fig.1:**

*An alternative to this careful editing of the mesh is the use of implicit schemes. However, using implicit schemes with CFL values much larger than 1 opens the door to both larger advection errors (stability does not imply accuracy) and larger splitting errors as the time steps for advection can be much smaller than the refraction and source term time step (Roland and Ardhuin 2014). We have preferred to stick to the explicit N-scheme because numerical efficiency is not central in the study, and it simplifies comparisons with global model results that also use explicit schemes. Implicit schemes are probably necessary when resolving regional scales and surf zones in the same mesh when CFL constraints require prohibitively small time steps in explicit schemes.*

OBS5. 63: Boulders of D50=15 cm. This is too small to be boulders, which are at least 25 cm, according to wikipedia. Cobbles, maybe?

**R5: The reviewer is correct. Based on the Wentworth grain size scale, sediments of D50 ~15cm are termed pebbles, thus bottom sediment size specifications in line 64 will be corrected as follows:**

*"...,while zones characterized as pebbles or larger elements (boulders) were represented with a D50=150 mm."*

OBS.6. 95: only tidal currents are used. Maybe it is OK to neglect general ocean currents, but it should be justified/explained here.

**R6: This is particularly an interesting point. Non-tidal flows are generally dominated by geostrophic currents. The impact of these currents are analyzed in (among others) Marechal and Ardhuin (2020), Alday et al. (2021) and Echevarria et al. (2021). Resolving the small scales of these currents at least 30 km is not feasible in a deterministic sense because the quasi-geostrophic dynamics are not yet resolved in observations (Ballarotta et al. 2019). Introducing modelled quasi-geostrophic currents at scales shorter than 100 km can often increase model errors (even though the result can be statistically better) because structures are not constrained by assimilation to be in the right place.**

OBS.7. 113: 30 sec time step is mentioned here, but we saw 13 sec limit earlier, so I don't understand the discrepancy. Please explain.

**R7: As explained in R3, the 13 s is obtained from the CFL condition for the lowest frequencies. The actual advection time step varies with spectral components and is 30 s at most.**

OBS.8. 125: maximum frequency of buoy should be mentioned. And later, the frequency range used for calculation of Tm02, etc.

**R8: The following specification will be added in the caption of Table 1:**

*"...Deploy depth obtained from model bathymetry interpolated into the buoys' position. All buoys with spectral data present a frequency range from 0.025 to 0.58 Hz, with a frequency interval of 0.005 Hz."*

**In relation to the wave parameters computed from the frequency spectrum, the following will be added to the caption of Fig.6:**

*"...Hs bin size is 0.25 m, periods' bin size is 0.2 s. Hs, Tm01 and Tm02 are computed integrating the spectra in the frequency range 0.0339-0.537 Hz."*

OBS.9. Section 4.1. It should be made more explicit up front here that the impact of resolution is studied by comparison against a global model, not by applying two regional models of different resolution. (Initially, I guessed the paper was doing the latter.) Same problem occurs on line 383.

**R9: In the case of section 4.1, we consider that the comparison method is properly explained from line 145 to line 148:**

**"A comparison between February 2011 mean significant wave heights (Hs) fields from the global model described in section 2.4 and our implemented regional model is presented in Fig. 5. To evaluate the differences between models, the output from the 0.5◦ global grid was linearly interpolated into the regional mesh nodes before computing the mean Hs and the NMD for the selected time window."**

**The text in lines 382 to 383 will be changed to:**

*"We found that higher spatial resolution adequate to solve bathymetry features and explicitly solving coastlines can introduce changes in Hs estimations of about 20% when compared to lower resolution global models"*

OBS.10. Figure 7. Is mean E(f) from an average over the 14 days? This should be specified.

**R10: All stats and mean E(f) in Fig. 7a-d are computed comparing 1 month simulations with buoy data. Time series presented in Fig. 7e-f are there to show the characteristics of H20 from simulations.**

**The following will be added in the caption of Fig.7:**

*"...In panels (a) to (d) 1 month modelled results compared with buoy data…"*

OBS.11. Section 4.4. I don't understand the difference between 48D24BC and 48D48BC. Both are interpolating from the same global model to the same regional model, right? Is it a question of doing the spectral interpolation in WW3 vs. outside WW3? This should be clearer.

**R11: To clarify the approach of the tests in Section 4.4 (specified in table 3), the following explanation will be added after the paragraph between lines 224 to 229:**

*"…For example, the difference between 48D24BC and 48D48BC is that the boundary conditions (BC) were created in a global model with different spectral resolution. Test 48D24BC employs boundary conditions from a global model with 24 discrete spectral directions equivalent to $15^0$ which are then interpolated into 48 directions to match the mesh resolution (48D). On the other hand, in test 48D48BC we used boundary conditions from a global model which has a spectral directional resolution of $7.5^0$ (48 directions) the same used in the high resolution mesh, hence, no directional interpolation of the spectrum is required."*

OBS12 Figure 17. I didn't realize until this figure (at the end of the paper) that the authors ran this regional model for a period of ~16 years. Was that mentioned anywhere? The comparisons prior to that are for much shorter periods, e.g. 1 month.

**R12: Indeed in Section 5 we used wave heights from a 28 years database created in the context of marine renewable energies. This database is described in Accensi et al. (2021) which can be found here: [https://archimer.ifremer.fr/doc/00736/84812/91664.pdf](https://archimer.ifremer.fr/doc/00736/84812/91664.pdf).**
**The use of the dataset is mentioned in line 346. The following will be added to make it more clear:**

*"...Making use of the coverage and improvements in this altimeter product, we analyzed the performance of our mesh over part of the wave hindcast described in Accensi et al. (2021), which was created using the same mesh employed in the present study."*

OBS13. 425: Is the satellite altimetry dataset mentioned here?

**R13: Thanks for pointing this out, the altimeter data set is in fact missing in the list. Options to access the ESA CCI V1 and V3 (same as v2 but open to public access) will be added in the Data Availability section:**

*"ESA Sea State CCI altimeter dataset access: [https://catalogue.ceda.ac.uk/](https://catalogue.ceda.ac.uk/) and [https://cersat.ifremer.fr/Data/Latest-products/](https://cersat.ifremer.fr/Data/Latest-products/)*

*Note that the CCI V2 altimeter dataset is not available to the public. Version 3 will be soon available to public access and is identical to V2 used in this study"*

---

## Author Comment (AC2)

**COMMENTS FROM REVIEWER #2:**

The authors investigate the impact on the model performance of modelling choices including spatial resolution, adjustments in wind-wave generation and swell dissipation, wave-current interactions, spectral resolution, bottom friction, forcing fields, and parameterizations of physical processes in a regional model that covers most of the Atlantic and North Sea coasts. The authors have presented a very nice and important guiding work that includes almost all the tests that need to be performed for the development of a nearshore wave model. The writing style and details are very well expressed. I think it can be accepted after very few minor corrections.

OBS.1 In the abstract: the fourth sentence of the abstract needs to be rephrased. More events are focused on in the study than are mentioned there. Also, the study does verification but this is not mentioned in the abstract.

**R1: The use of buoy and satellite data to analyze the model output will be included in the abstract. The performance analysis approach used in Section 5 will also be mentioned here to cover the different elements included in this study.**

**The abstract will be modified as follows:**

*"Numerical wave models are generally less accurate in the coastal ocean than offshore. It is generally suspected that a number of factors specific to coastal environments can be blamed for these larger model errors: complex shoreline and topography, relatively short fetches, combination of remote swells and local wind seas, less accurate wind fields, presence of strong currents, bottom friction, etc. These factors generally have strong local variations, making it all the more difficult to adapt a particular model setup from one area to another. Here we investigate a wide range of modelling choices including forcing fields, spectral and spatial resolution, and parameterizations of physical processes in a regional model that covers most of the Atlantic and North Sea coasts. The effects of these choices on the model results are analyzed with buoy spectral data and wave parameters' time series. Additionally, satellite altimeter data is employed to provide a more complete performance assessment of the modelled wave heights as function of the distance to the coast and to identify areas where wave propagation is influenced by bottom friction. We show that the accurate propagation of waves from offshore is probably the most important factor on exposed shorelines, while other specific effects can be important locally, including winds, currents and bottom friction."*

OBS.2 In line 37: "section 3" should be.

**R2: Phrase in line 37 has been changed to :**

*"...Wave measurements used for sensitivity analyses and validation in section 3…"*

OBS.3 The title of chapter 4 should be "model performance indicators" and should be separated from the following sub-sections. The sub-headings that appear under the title of chapter 4 should be given under the main title of results and discussion.

**R3: Chapter 4 has been renamed as "Model performance indicators" and a new Chapter 5 has been created entitled "Sensitivity analyses results and discussion" which contains the previous sub-sections from Chapter 4.**

OBS.4  In line 169: "Ponce de Leon" must be.

**R4: Thanks for noting this. The bibtex file will be corrected to properly display the name of the cited author.**

OBS.5 In Fig. 9: "Sgnificant" must be "Significant"

**R5: Misspelling in Fig.9a has been corrected to "Significant wave height"**

OBS.6 In line 232: "were presented in Fig. 4" must be.

**R6: Phrasing in line 232 was changed to :**

*"Bathymetry details of these locations were presented in Fig. 4."*

OBS.7 In line 309: "altimeters" must be.

**R7: Misspelling in line 309 was corrected to "altimeters".**

OBS.8 In the title of Fig. 14: please check this word "mayor"

**R8: Thanks for pointing this out. Phrasing in the caption of Fig.14 was changed to:**

*" Magenta ovals in (b) highlight areas with largest bias reduction"*

---

## Author Response (AR1)

Dear Reviewers,

On behalf of all the Co-Authors I would like to thank you for your comments and insights, and for pointing out those details that helped to improve the quality of the manuscript (in content and format). Below you will find our responses and changes applied to the document. Note that all identified lines where corrections/modifications were introduced, correspond to the new version of the document with "track changes".

Best regards,

 Matias Alday

**COMMENTS FROM REVIEWER #1:**

Review of "Accuracy of numerical wave model results: Application to the Atlantic coasts of Europe"

a paper by Alday and 3 co-authors.

Summary:

The authors design an unstructured grid implementation of WW3 for the European coastline. They evaluate the model against buoys and altimetry. They perform sensitivity analyses to determine impact of various things. Topics studied include: simple wind adjustments, swell dissipation settings, tidal currents effects, directional resolution, and bottom friction sensitivity.

Recommendation:

I'm not very familiar with the expectations of the journal, especially re: what types of papers are OK. I have a generally positive impression of the paper. So, I'd like to give the authors the benefit of the doubt, and recommend "accept with minor revisions"

**General comments.**

Some of the comparisons seem rather workaday, like the sort of thing that would go into a tech report or dissertation. But it's OK, I think. And they are not just dry comparisons: the authors put substantial thought into them, so nothing comes across as especially unnecessary.

The literature review is sufficient.

The comparisons are careful, the discussion is straightforward/honest, and the authors don't over-sell the outcomes.

The findings are potentially useful for people doing similar modeling.

The use of English is very good. There are a few awkward or broken sentences (e.g. lines 310, 393-394) and minor style errors (e.g. lines 21, 27, 409, 412), but not more than ~ 1 problem per page.

The purpose of the study meanders.

There are a lot of evaluations, but they are a grab-bag that are only similar insofar as they are for the same system. I put them in four broad categories:

1) differences vs. global model (geographic resolution)

2) impact of design choices, where choice is not clear a priori (swell dissipation setting, directional resolution, BC formatting, wind adjustments)

3) impact of including/excluding effect, where it is more or less accepted a priori that the effect should be included (bottom friction, tidal current)

4) general evaluation of the model in its final form (altimetry comparison).

I don't have a good suggestion for how to address this. I suppose the authors do not want to stick to a consistent type of content, since they would lose half the study.

I really like the evaluation of the model against altimetry, sorted by distance from the coast. Excellent.

**COMMENT 1. There are a number of unexplained acronyms/initialisms. Perhaps a glossary would work better than rigorous in-line explanation. I leave this to the editor to decide.**

**R1**: We have significantly reduced the use of acronyms in order to improve readability of the manuscript.

**Changes applied 1**:

- the number of occurrences of "NMD" (normalized mean difference) has been reduced from 16 to 5
- the 6 occurrences of WAE (Wave Action Equation) have been removed,
- The 3 occurrences of MSL have been removed
- 10 occurrences of NMB have been removed
- 10 occurrences of SI have been removed
- 20 occurrences of SWH or $H_s$ have been removed

**COMMENT 2. When printed out, some of the figures are hard to see, like Fig 7 a-d, where lines are hard to distinguish.**

**R2:** Some figures were indeed a bit busy. We have chosen to simplify the figures to make it possible to magnify them.

**Changes applied 2:** In the case of Figure 7 the normalized bias plots were removed.

**Specific comments.**

**COMMENT 3. "CFL...minimum time step". Wouldn't CFL dictate the maximum time step?**

**R3:** Indeed the actual time step used at run time is automatically adjusted between a maximum value and whatever it takes to keep the CFL number under 1, and this adjustment is done separately for each spectral component (higher frequencies that propagate slower are typically integrated with higher time steps).

**Changes applied 3:** This sentence, now between lines 54 to 57 in the document with track changes, was rewritten for clarity:

*"This coarsening allowed a lower Courant-Friedrich-Lewy (CFL) number, which makes it possible to use larger time step for wave propagation, 13~s in this case for our lowest frequencies, but it also implies that details of the Norwegian coastline are not as well resolved".*

**COMMENT 4. Same paragraph. Why is the implicit scheme not used? It should be mentioned, at least.**

**R4:** We have added an explanatory paragraph after Fig.1

**Changes applied 4:** The following was added between lines 63 to 68 in the document with track changes:

*"An alternative to this careful editing of the mesh is the use of implicit schemes. However, using implicit schemes with CFL values much larger than 1 opens the door to both larger advection errors (stability does not imply accuracy) and larger splitting errors as the time steps for advection can be much smaller than the refraction and source term time step (Roland and Ardhuin 2014). We have preferred to stick to the explicit N-scheme because numerical efficiency is not central in the study, and it simplifies comparisons with global model results that also use explicit schemes. Implicit schemes are probably necessary when resolving regional scales and surf zones in the same mesh when CFL constraints require prohibitively small time steps in explicit schemes."*

**COMMENT 5. Boulders of D50=15 cm. This is too small to be boulders, which are at least 25 cm, according to wikipedia. Cobbles, maybe?**

**R5:** The reviewer is correct. Based on the Wentworth grain size scale, sediments of $D_{50}$ ~15cm are termed pebbles, thus bottom sediment size specifications in (originally) line 64 will be corrected.

**Changes applied 5:** Phrase between lines 73 to 74 (document with track changes) has been changed to:

*"...,while zones characterized as pebbles or larger elements (boulders) were represented with a $D_{50}$=150 mm."*

**COMMENT 6. Only tidal currents are used. Maybe it is OK to neglect general ocean currents, but it should be justified/explained here.**

**R6:** This is particularly an interesting point. Non-tidal flows are generally dominated by geostrophic currents. The impact of these currents are analyzed in (among others) Marechal and Ardhuin (2020), Alday et al. (2021) and Echevarria et al. (2021). Resolving the small scales of these currents at least 30 km is not feasible in a deterministic sense because the quasi-geostrophic dynamics are not yet resolved in observations (Ballarotta et al. 2019). Introducing modelled quasi-geostrophic currents at scales shorter than 100 km can often increase model errors (even though the result can be statistically better) because structures are not constrained by assimilation to be in the right place.

**Changes applied 6:** The following was specified between lines 200 to 202 (tack changes document):

*"...although a full effect generally requires relatively high spatial resolution that is generally not achievable by observations and thus models are usually not constrained at the necessary scale (Marechal and Ardhuin, 2020). This is the main reason why geostrophic currents were not considered in the high resolution regional model."*

**COMMENT 7. 113: 30 sec time step is mentioned here, but we saw 13 sec limit earlier, so I don't understand the discrepancy. Please explain.**

**R7:** As explained in R3, the 13 s is obtained from the CFL condition for the lowest frequencies. The actual advection time step varies with spectral components and is 30 s at most.

**Changes applied 7:** None.

**COMMENT 8. 125: maximum frequency of buoy should be mentioned. And later, the frequency range used for calculation of Tm02, etc.**

**R8:** Good point here. This will provide more clarity on the used data and the computed parameters.

**Changes applied 8:** The following specification will be added in the caption of Table 1:

*"...Deploy depth obtained from model bathymetry interpolated into the buoys' position. All buoys with spectral data present a frequency range from 0.025 to 0.58 Hz, with a frequency interval of 0.005 Hz."*

In relation to the wave parameters computed from the frequency spectrum, the following was added to the caption of Fig.6:

*"...Hs bin size is 0.25 m, periods' bin size is 0.2 s. Hs, Tm01 and Tm02 are computed integrating the spectra in the frequency range 0.0339-0.537 Hz."*

**COMMENT 9. Section 4.1. It should be made more explicit up front here that the impact of resolution is studied by comparison against a global model, not by applying two regional models of different resolution. (Initially, I guessed the paper was doing the latter.) Same problem occurs on line 383.**

**R9:** In the case of section 4.1 (now section 5.1), we consider that the comparison method is properly explained from line 159 to line 164 (track changes document):

"A comparison between February 2011 mean Hs fields from the global model described in section 2.4 and our implemented regional model is presented in Fig. 5. To evaluate the differences between models, the output from the 0.5◦ global grid was linearly interpolated onto the regional mesh nodes before computing the mean wave height and the mean difference for the selected time window."

In the conclusions we will clarify that we are talking about comparisons with global models

**Changes applied 9:** The text in lines 411 to 413 was changed to:

*"We found that higher spatial resolution adequate to solve bathymetry features and explicitly solving coastlines can introduce changes in modelled wave height estimations of about 20 % when compared to lower resolution global models."*

**COMMENT 10. Figure 7. Is mean E(f) from an average over the 14 days? This should be specified.**

**R10:** All stats and mean E(f) in Fig. 7a-d are computed comparing 1 month simulations with buoy data. Time series presented in Fig. 7e-f are there to show the characteristics of H20 from simulations.

**Changes applied 10:** The following will be added in the caption of Fig.7:

*"...In panels (a) to (d) 1 month modelled results compared with buoy data…"*

**COMMENT 11. Section 4.4. I don't understand the difference between 48D24BC and 48D48BC. Both are interpolating from the same global model to the same regional model, right? Is it a question of doing the spectral interpolation in WW3 vs. outside WW3? This should be clearer.**

**R11:** Further details will be added to clarify the analysis approach from tests in Section 4.4 (now Section 5.4) .

**Changes applied 11:** To clarify the approach of the tests in Section 5.4 (specified in table 3), the following explanation was added after the paragraph between lines 245 to 250 (track changes document):

*"…For example, the difference between 48D24BC and 48D48BC is that the boundary conditions (BC) were created in a global model with different spectral resolution. Test 48D24BC employs boundary conditions from a global model with 24 discrete spectral directions equivalent to 15⁰ which are then interpolated into 48 directions to match the mesh resolution (48D). On the other hand, in test 48D48BC we used boundary conditions from a*

*global model which has a spectral directional resolution of 7.5⁰ (48 directions) the same used in the high resolution mesh, hence, no directional interpolation of the spectrum is required."*

**COMMENT 12.** Figure 17. I didn't realize until this figure (at the end of the paper) that the authors ran this regional model for a period of ~16 years. Was that mentioned anywhere? The comparisons prior to that are for much shorter periods, e.g. 1 month.

**R12:** Indeed in Section 5 we used  wave heights from a 28 years database created in the context of marine renewable energies. This database is described in Accensi et al. (2021) which can be found here: https://archimer.ifremer.fr/doc/00736/84812/91664.pdf.

The use of the dataset is mentioned in line 346 of the original document.

**Changes applied 12:**  The following was included to make it more clear (lines 374 to 376 of the updated document with track changes):

*"...Making use of the coverage and improvements in this altimeter product, we analyzed the performance of our mesh over part of the wave hindcast described in Accensi et al. (2021), which was created using the same mesh employed in the present study."*

**COMMENT 13.** 425: Is the satellite altimetry dataset mentioned here?

**R13:** Thanks for pointing this out, the altimeter data set is in fact missing in the list.

**Changes applied 13:** Options to access the ESA CCI V1 and V3 (same as v2 but open to public access) were added in the Data Availability section:

*"ESA Sea State CCI altimeter dataset access: https://catalogue.ceda.ac.uk/ and https://cersat.ifremer.fr/Data/Latest-products/*

*Note that the CCI V2 altimeter dataset is not available to the public. Version 3 will be soon available to public access and is identical to V2 used in this study"*

**COMMENTS FROM REVIEWER #2:**

The authors investigate the impact on the model performance of modelling choices including spatial resolution, adjustments in wind-wave generation and swell dissipation, wave-current interactions, spectral resolution, bottom friction, forcing fields, and parameterizations of physical processes in a regional model that covers most of the Atlantic and North Sea coasts. The authors have presented a very nice and important guiding work that includes almost all the tests that need to be performed for the development of a nearshore wave model. The writing style and details are very well expressed. I think it can be accepted after very few minor corrections.

**COMMENT 14. In the abstract: the fourth sentence of the abstract needs to be rephrased. More events are focused on in the study than are mentioned there. Also, the study does verification but this is not mentioned in the abstract.**

**R14:** The use of buoy and satellite data to analyze the model output will be included in the abstract. The performance analysis approach used in Section 5 will also be mentioned here to cover the different elements included in this study.

**Changes applied 14:** The abstract was modified as follows

*"Numerical wave models are generally less accurate in the coastal ocean than offshore. It is generally suspected that a number of factors specific to coastal environments can be blamed for these larger model errors: complex shoreline and topography, relatively short fetches, combination of remote swells and local wind seas, less accurate wind fields, presence of strong currents, bottom friction, etc. These factors generally have strong local variations, making it all the more difficult to adapt a particular model setup from one area to another. Here we investigate a wide range of modelling choices including forcing fields, spectral and spatial resolution, and parameterizations of physical processes in a regional model that covers most of the Atlantic and North Sea coasts. The effects of these choices on the model results are analyzed with buoy spectral data and wave parameters' time series. Additionally, satellite altimeter data is employed to provide a more complete performance assessment of the modelled wave heights as function of the distance to the coast and to identify areas where wave propagation is influenced by bottom friction. We show that the accurate propagation of waves from offshore is probably the most important factor on exposed shorelines, while other specific effects can be important locally, including winds, currents and bottom friction."*

**COMMENT 15. In line 37: "section 3" should be.**

**R15:** Missing word added.

**Changes applied 15:** Phrase in line 39-40 was changed to

*"...Wave measurements used for sensitivity analyses and validation in section 3…"*

**COMMENT 16. The title of chapter 4 should be "model performance indicators" and should be separated from the following sub-sections. The sub-headings that appear under the title of chapter 4 should be given under the main title of results and discussion.**

**R16:** This is a good suggestion in terms of format.

**Changes applied 16:** Chapter 4 has been renamed as "Model performance indicators" and a new Chapter 5 has been created entitled "Sensitivity analyses results and discussion" which contains the previous sub-sections from Chapter 4.

**COMMENT 17.** In line 169: **"Ponce de Leon" must be**.

**R17:** Thanks for noting this.

**Changes applied 17:** The bibtex file was corrected to properly display the name of the cited author.

**COMMENT 18. In Fig. 9: "Sgnificant" must be "Significant"**

**Changes applied 18:** Misspelling in Fig.9a has been corrected to "Significant wave height"

**COMMENT 19. In line 232: "were presented in Fig. 4" must be.**

**Changes applied 19:** Phrasing in current line 255 was changed to :

*"Bathymetry details of these locations were presented in Fig. 4."*

**COMMENT 20. In line 309: "altimeters" must be.**

**Changes applied 20:** Misspelling in line 355 (in the tack changes document) was corrected to "altimeters".

**COMMENT 21. In the title of Fig. 14: please check this word "mayor"**

**R21:** Thanks for pointing this out. The expression will be corrected.

**Changes applied 21:** Phrasing in the caption of Fig.14 was changed to:

*" Magenta ovals in (b) highlight areas with largest bias reduction."*